# Accelerated deprotonation with a hydroxy-silicon alkali solid for rechargeable zinc-air batteries

Yaobin Wang[1], Xinlei Ge [1], Qian Lu [1] ✉, Wenjun Bai[2], Caichao Ye [2] ✉, Zongping Shao[3] ✉ & Yunfei Bu [1] ✉

Transition metal oxides are promising electrocatalysts for zinc-air batteries, yet surface reconstruction caused by the adsorbate evolution mechanism, which induces zinc-ion battery behavior in the oxygen evolution reaction, leads to poor cycling performance. In this study, we propose a lattice oxygen mechanism involving proton acceptors to overcome the poor performance of the battery in the OER process. We introduce a stable solid base, hydroxy BaCaSiO$_4$, onto the surfaces of PrBa$_{0.5}$Ca$_{0.5}$Co$_2$O$_{5+\delta}$ perovskite nanofibers with a one-step exsolution strategy. The HO-Si sites on the hydroxy BaCaSiO$_4$ significantly accelerate proton transfer from the OH* adsorbed on PrBa$_{0.5}$Ca$_{0.5}$Co$_2$O$_{5+\delta}$ during the OER process. As a proof of concept, a rechargeable zinc-air battery assembled with this composite electrocatalyst is stable in an alkaline environment for over 150 hours at 5 mA cm$^{-2}$ during galvanostatic charge/discharge tests. Our findings open new avenues for designing efficient OER electrocatalysts for rechargeable zinc-air batteries.

Rechargeable Zn-air batteries have attracted great attention owing to their high energy densities, environmental friendliness, and low costs, in which the oxygen evolution reaction (OER) determines the charging efficiency[1–4]. Recently, the development of efficient oxygen electrocatalysts has been seen as pivotal in realizing practical Zn-air batteries. Transition metal oxides are important oxygen electrocatalysts. However, the conventional adsorbate evolution mechanism (AEM) induces a valence change in the transition metal and even surface reconstruction during the OER, which results in behaviors typical of zinc-ion batteries, specifically, the emergence of two voltage platforms. To address this issue, the precise design of advanced oxygen electrocatalysts for rechargeable Zn-air batteries is imperative.

Recently, the lattice oxygen mechanism (LOM) was introduced to break the linear relationship and limit the theoretical overpotential for AEMs. In the LOM, the catalytic sites are the oxygen sites, and the transition metal does not participate in the reaction. Given this, we believe that introducing LOM into Zn-air batteries could improve the zinc-ion battery behavior. However, the OER via LOM exhibited poor stability and a high proton desorption kinetic barrier. Therefore, it is essential to modify oxygen electrocatalysts that utilize the LOM to improve their activities and stabilities. Various oxygen electrocatalysts have garnered significant interest due to their adjustable components, tunable electronic structures, and competitive performance compared to noble metals[5,6]. Most research on perovskite oxides is designed to optimize their electronic configurations to achieve favorable activities[7]. For example, the Shao-Horn group discovered that the OER catalytic activity of perovskite oxides largely depended on the energetics of charge transfer in four proton-coupled electron transfer (PCET) processes[8,9]. Specifically, the deprotonation process, which converts OH* to O*, was identified as the rate-determining step for

[1]Jiangsu Key Laboratory of Atmospheric Environment Monitoring and Pollution Control (AEMPC), Jiangsu Collaborative Innovation Center of Atmospheric Environment and Equipment Technology, UNIST-NUIST Energy and Environment Jointed Lab, (UNNU), School of Environmental Science and Technology, Nanjing University of Information Science and Technology (NUIST), 219 Ningliu, Nanjing 210044, P. R. China. [2]Academy for Advanced Interdisciplinary Studies & Department of Materials Science and Engineering, Southern University of Science and Technology, Shenzhen 518055, China. [3]WA School of Mines: Minerals, Energy and Chemical Engineering (WASM-MECE), Curtin University, Perth, WA 6845, Australia. ✉ e-mail: luqian_0104@nuist.edu.cn; yecc@sustech.edu.cn; shaozp@njtech.edu.cn; yunfei.bu@nuist.edu.cn

highly covalent perovskite oxides. Building on this, the Shao group developed a composite catalyst with $Sr_3B_2O_6$ as a proton acceptor supported on the surface of $SrCo_{0.8}Fe_{0.2}O_{3-\delta}$[10]. The O $2p$ band center was close to the Fermi energy, suggesting that lattice oxygen was involved in the OER process and that deprotonation of OH* or OOH* was the rate-limiting step[8]. This enhancement in proton transfer at the interface significantly increased the OER catalytic activity.

The slow deprotonation rate in the OER is attributed to the low energy of the lone pair electrons surrounding the oxygen atoms, which results in weak binding between the oxygen atom and the proton[11]. To address this challenge, Ciucci et al. introduced a $MoS_2@SrCoO_{3-\delta}$ perovskite to enhance electron-proton transfer, which overcame the proton-transfer limit. The interfacial $MoS_2$ served as both an electron and proton acceptor in this system. However, the origin of the improved OER activity, which may have arisen from enhanced proton transport or electronic structure changes in the perovskite oxide, remains uncertain[12,13]. The roles of proton acceptors in catalytic reactions is also a subject of ongoing debate[14]. In light of this, we suggest a surface phosphate modification strategy that does not alter the electronic structure of the perovskite oxide. Molecular-level phosphate can serve as a proton acceptor to enhance the catalytic activity of a perovskite oxide[15]. However, the electrochemical stabilities of proton acceptors and the interface can influence the durability of the perovskite oxide and significantly impact the cycling lives of Zn-air batteries. Establishing a stable interface between a robust proton acceptor and a perovskite oxide remains a formidable challenge.

Reports indicated that silicon compounds exhibited excellent electrochemical stability in alkaline media and have the optimal Gibbs free energy for H adsorption[16], making them ideal proton acceptors for efficient H adsorption/desorption. In our study, we incorporated $BaCaSiO_4$ (BCS) nanoparticles onto the surface of $PrBa_{0.5}Ca_{0.5}Co_2O_{5+\delta}$ (PBCC) with a simple one-step exsolution method in which tetraethyl orthosilicate was the silicon source. BCS-PBCC showed a low overpotential of 300 mV at 10 mA cm$^{-2}$ in 1.0 M KOH. There was a strong correlation between the amount of incorporated Si and the Tafel slope, with the Tafel slope decreasing as the amount of Si incorporated increased (1–5 wt%). We quantified the participation ratio of the lattice oxygens in the OER with in situ differential electrochemical mass spectrometry (DEMS) measurements to confirm the LOM operated with the BCS-PBCC catalyst. Density functional theory calculations revealed an LOM involving proton transfer at the interface between BCS and PBCC. The unique hydroxy silicon configuration (HO-Si-O-$Ca^{2+}$/$Ba^{2+}$-O-Si-OH) facilitated the deprotonation of adsorbed OH* on the perovskite surface. As a proof of concept, Zn-air batteries assembled with BCS-PBCC exhibited a low charging voltage of 1.93 V at 50 mA cm$^{-2}$ and a stable charging voltage at 5 mA cm$^{-2}$ for over 150 hours. The use of robust hydroxy silicon as a proton acceptor could pave the way for designing efficient OER electrocatalysts for rechargeable Zn-air batteries or water-splitting devices.

## Results

### Synthesis and structural characterization of BCS-PBCC

Both BCS-PBCC and PBCC were produced with the electrospinning technique (Fig. 1a, Methods section). At high calcination temperatures, the perovskite parent maintained a stable structure with hexacoordinated cobalt in the B-sites. As a result, the four-coordinated silicon was readily exsolved onto the surface of PBCC during calcination at 950 °C (Fig. 1b)[17]. The characteristic diffraction peaks for BCS at 30° and 32° were clearly observed in the X-ray diffraction (XRD) patterns of BCS$_x$-PBCC with various doping levels, as shown in Figs. S1 and S2. In addition, the crystal structure of PBCC remained intact after BCS doping. The refined XRD data reveal that the PBCC perovskite adopted a layered perovskite structure with space group P4/mmm and lattice parameters of a = b = 3.900 Å and c = 7.618 Å. Conversely, $BaCaSiO_4$ adopted a cubic structure with a space group of P63/mm and lattice

parameters of a = b = 5.751 Å and c = 14.671 Å, as shown in Fig. 1c (Supplementary Table 1).

The BCS nanoparticles were uniformly distributed on the surfaces of the PBCC fibers to prepare various doped electrocatalysts, as illustrated in Fig. S3a–c. Notably, the density of BCS nanoparticles on the surface increased with increasing doping level ($x$ in BCS$_x$-PBCC) from 0.01 to 0.2. The positive correlation between the density of BCS and the doping level confirmed that the formation of high-valent $SiO_4^{4-}$ and migration of BCS to the PBCC surface was thermodynamically spontaneous when tetraethyl orthosilicate was used as the silicon source in the precursor solution. The energy dispersive X-ray spectroscopy (EDX) analysis shown in Fig. S4 confirmed that the BCS nanoparticles were well dispersed on the PBCC surface. The particles of BCS measured just 10 nm, as shown in Fig. 1d, and the average diameter of the PBCC nanofibers was ~500 nm; therefore, a well-defined surface-supported structure was obtained, as shown in Fig. S5. The heterostructures between the BCS and PBCC were examined with high-resolution transmission electron microscopy (HRTEM). The BCS nanoparticles were tightly anchored on the PBCC surface with a clear heterointerface, as shown in Fig. 1e. This was significantly different from the original sample of PBCC (Figs. S6 and S7). In addition, the (110) plane in the crystalline lattice of BCS coincided with the (100) plane of PBCC, as shown in Fig. 1f. This confirmed the exsolution process of the BCS nanoparticles. The FFT pattern also revealed the atomic distribution in the intertwined region where BCS and PBCC overlapped along the axial direction.

### Comprehensive evaluation of the OER performance

The OER activity was assessed with a three-electrode configuration and a rotating disk electrode (RDE under ambient conditions). Figure 2a presents the iR-corrected polarization curves for a series of BCS$_x$-PBCC electrocatalysts in $O_2$-saturated 1.0 M KOH solutions, with all potentials referenced to the reversible hydrogen electrode (RHE). The results indicated that the BCS-modified PBCC displayed enhanced OER catalytic activity, with a larger current density and lower overpotential than pristine PBCC. Significantly, the BCS-PBCC catalyst with 5 wt% silicon doping exhibited an onset potential of 1.45 V and an overpotential ($\eta$) of only 300 mV at an anodic current density of 10 mA cm$^{-2}$. This constituted a considerable advancement relative to pristine PBCC, which had an overpotential of 415 mV. The pure BCS showed poorer OER activity than BCS-PBCC and PBCC, which meant that the PBCC provided the active sites in the hybrid BCS-PBCC electrocatalyst (Fig. S8). Excess doped silicon covered the active crystal planes of PBCC and decreased the number of catalytic sites, thus decreasing the overall catalytic performance. A comparison between physically mixed catalysts (5% BCS + PBCC) and BCS-PBCC prepared by in situ exsolution showed that an intimate interface between the BCS and PBCC was the decisive factor for the OER activity (Fig. S9). Conversely, the sizes of the large BCS particles could impede their participation in the OER process of PBCC, especially when taking into account the geometric arrangement of BCS and PBCC.

The Tafel plot, which was derived from the polarization curves, is shown in Fig. 2b, and it sheds light on the OER process and dynamics. The Tafel slope for the pristine PBCC catalyst was 85 mV dec$^{-1}$. This suggested that the OER process with PBCC was rate-limited in the second stage, in which the active sites formed strong bonds with OH groups and deprotonation of OH* was hardly possible. However, introduction of the proton acceptor BCS dramatically decreased the Tafel slopes of BCS$_x$-PBCC ($x$ = 0.01, 0.025, and 0.05) hybrids to 71, 63, and 49 mV dec$^{-1}$, respectively. These values were notably lower than that of PBCC and approached the characteristic Tafel slope of 85 mV dec$^{-1}$. These observations confirmed that integrating the proton acceptor BCS fundamentally altered the potential-determining step of the OER, thereby accelerating deprotonation of the reactants.

Figure 2c, d illustrates the mass activity (MA) and specific activity (SA) of the BCS$_x$-PBCC electrocatalyst, respectively, at a normalized overpotential of $\eta = 320$ mV. While the specific activity was calculated based on the specific surface area (Fig. S10a, b), the mass activity was determined based on the oxide mass loading. When evaluated in relation to the electrochemical surface area (ECSA), the catalytic activity showed the same trend as the SA (Fig. 2d and S11). To mitigate the influence of the specific surface area on the performance, we also analyzed the intrinsic activity (Fig. S12). The turnover frequency (TOF), denoting the quantity of gaseous hydrogen molecules released per second per active oxygen site, is another crucial parameter reflecting the intrinsic activity of an electrocatalyst. Figure 2e reveals that proper doping significantly enhanced the intrinsic activity of the catalyst. The OER performance studies demonstrated that the BCS-PBCC electrocatalyst had excellent OER catalytic activity, surpassing those of most well-known perovskite-type OER electrocatalysts (Fig. 2f, Supplementary Table 2)[18].

The notable potential of the BCS-PBCC for use as an OER catalyst prompted the construction of an electrocatalytic cell comprising a commercial Pt/C cathode and the BCS-PBCC electrocatalyst as the anode (denoted as BCS-PBCC || Pt/C) for overall water splitting in a 1.0 M KOH electrolyte solution (Fig. S13). The long-term stability of this two-electrode battery was evaluated with chronopotentiometry. As shown in Fig. S14a–c, BCS-PBCC || Pt/C produced nearly stable potentials during continuous production of oxygen and hydrogen for 100 hours at a constant current density of 10 mA cm$^{-2}$. The structural and compositional changes of BCS-PBCC were also analyzed after an accelerated durability test (ADT), and the XRD pattern remained unchanged after more than 100 cycles (Fig. S15). Further analysis, as shown in Fig. S16a–c, confirmed that the BCS nanoparticles remained evenly dispersed across the perovskite matrix after the reaction. This uniform distribution was attributed to robust binding resulting from self-growth of the heterostructure. These observations attest to the advantages of the one-step in situ exsolution synthetic approach, which yielded a superior and enduring BCS-PBCC structure and outclassed prior methodologies.

## Origin of the improved OER activity

To elucidate the source of the OER activity, the effects of Si incorporation on the physicochemical properties were analyzed. In the formation of BCS proton acceptors, the proton serves as a Lewis acid, because it attaches to an electron donor[19]. For example, the reaction of

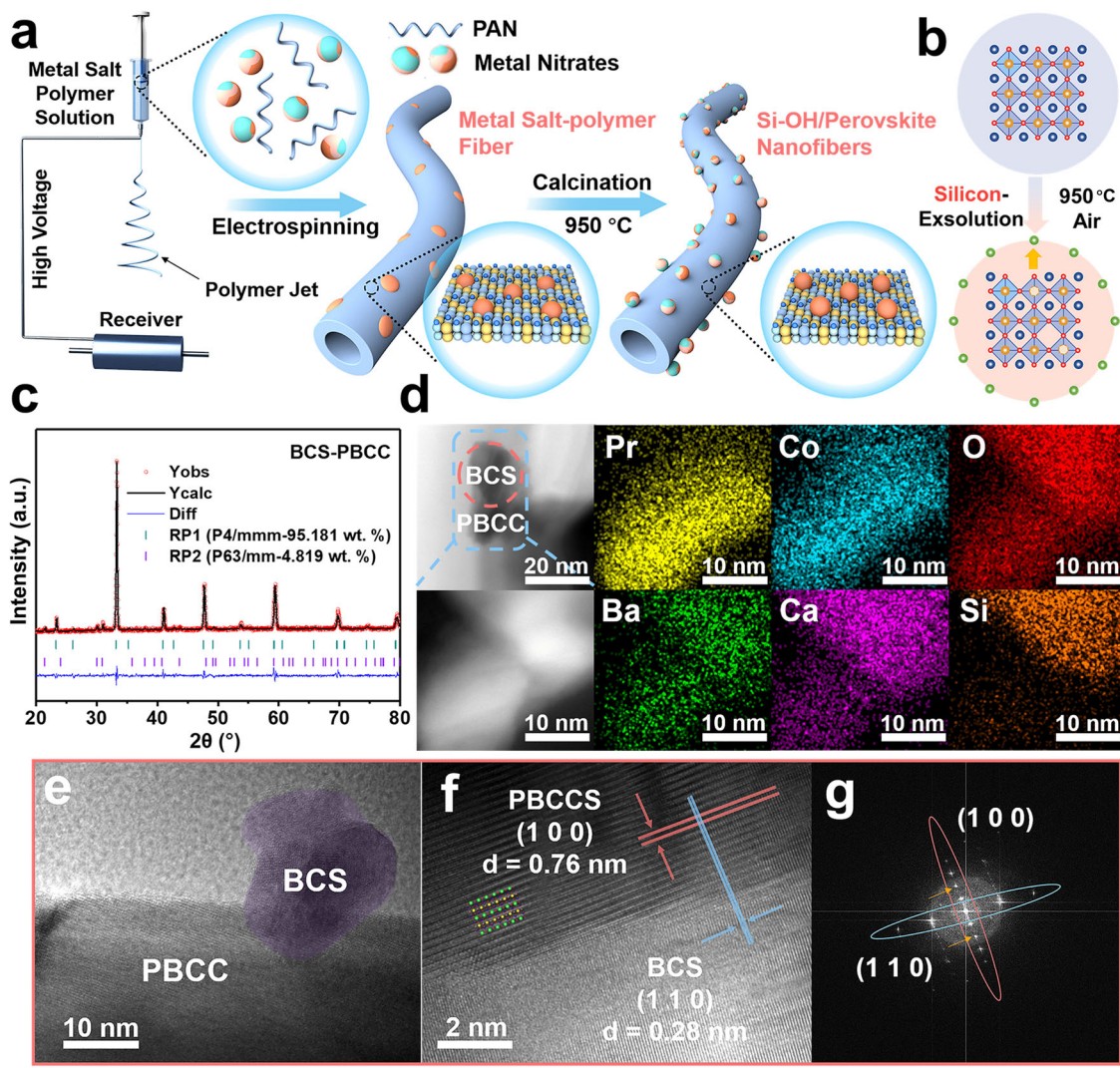

**Fig. 1 | Preparation and characterization of BCS-PBCC. a** Schematic illustration showing the procedure used to prepare BCS-PBCC. Color representation: Pr (yellow), Ba (blue), Ca (light blue), Co (navy blue), Si (orange). **b** Silicon evolution from perovskite oxide structures. Color representation: Pr/Ba/Ca (blue), Co (yellow), Si (green), O (red). **c** Refined XRD profile for BCS-PBCC. **d** HAADF-STEM image and the corresponding elemental maps for BCS-PBCC. **e** TEM and **f** HRTEM images of BCS-PBCC and **g** the corresponding FFT pattern for BCS-PBCC.

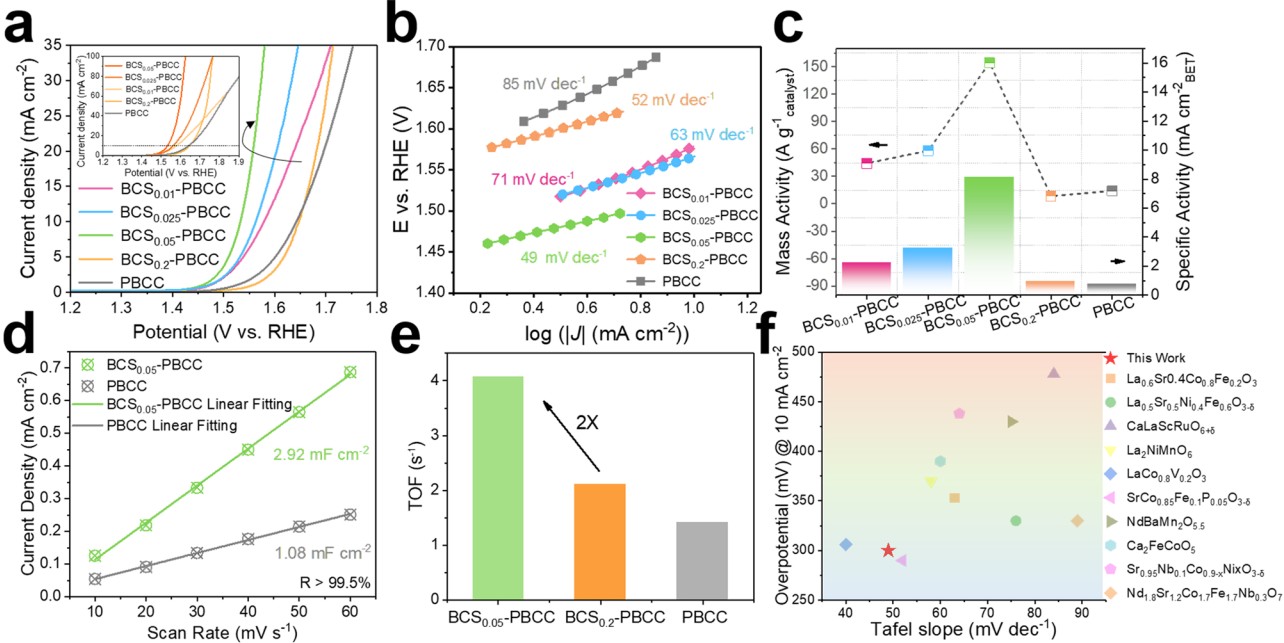

**Fig. 2 | Electrocatalytic OER performance in alkaline media. a** Polarization curves for BCS$_{0.01}$-PBCC, BCS$_{0.025}$-PBCC, BCS$_{0.05}$-PBCC and the original PBCC catalyst in O$_2$-saturated 1.0 M KOH solutions. The inset shows the polarization curve at a high current density. The scan rate was 5 mV s$^{-1}$. **b** The corresponding Tafel plots. **c** Normalized specific activity based on the mass and Brunauer–Emmett–Teller surface area of the electrocatalysts at $\eta = 1.53$ V. **d** Corresponding linear fit of the capacitive current to the CV scan rate. **e** Turnover frequency of BCS$_{0.05}$-PBCC, BCS$_{0.2}$-PBCC and PBCC. **f** OER activity comparison for BCS-PBCC and reported state-of-the-art perovskite oxide catalysts.

Ca$^{2+}$/Ba$^{2+}$ with SiO$_2$ to form polyanionic [SiO$_4^{4-}$] silicate can be interpreted as the transfer of basic O$^{2-}$ ions from the weakly acidic Ca$^{2+}$/Ba$^{2+}$ to the strongly acidic Si$^{4+}$. The increased electron density on the oxygen atoms thus increased the ability to form H bonds, resulting in stabilization of the coordinated H = O (Si) by additional hydrogen bonds and altered geometry (Si-O-Si bond angles deviate from 180°)[14]. The formation of two H-bonds provided exceptional stability and indicated significant electron-pair donor-proton acceptor capability (solid base). Stable acceptor structures that form OH-Si-O-Si-OH or OH-Si-O-Ca$^{2+}$/Ba$^{2+}$-Si-OH species are more likely to accept protons from the intermediate OH*, resulting in a 10-fold enhancement in the overall OER performance (Fig. 3a)[20].

To elucidate the enhanced OER activity attributed to BCS addition and to determine the mechanism for proton acceptor regulation of the PBCC perovskite, a series of BCS$_x$-PBCC samples was extensively characterized. First, the Co oxidation states were probed with X-ray photoelectron spectroscopy (XPS), since it is commonly recognized that the B sites in the perovskite (occupied by Co) are the active catalytic centers (as illustrated in Fig. 3b). A lower Co oxidation state is correlated with the formation of oxygen vacancies. Notably, BCS-PBCC exhibited a lower average valence for the host Co cations (+3.52 → +3.39) in comparison with PBCC. Furthermore, BCS-PBCC had more O$_2^{2-}$/O$^-$ species on its surface than PBCC, suggesting more surface oxygen vacancies, which was reported to be beneficial for the OER. The O 1s XPS results revealed a marginal increase in OH species, which may have stemmed from the adsorption of hydroxy groups by the surface proton acceptors (as depicted in Fig. 3c)[21]. These results shed light on the mechanism for the improved performance.

Raman spectroscopic measurements provided further evidence for the formation of hydroxy silicon and shed light on bonding of the Si in the structure (as illustrated in Fig. 3d)[22]. The 332 cm$^{-1}$ Raman peak was associated with Co-O-Co bending, and the 495 cm$^{-1}$ Raman peak corresponded to the Co-O-Si asymmetric stretching mode, highlighting the connection of the BCS to the PBCC structure. A band at 950 cm$^{-1}$ was seen for BCS-PBCC, which was linked to Si-O-Si network

bending and Si-O-Ca-O-Si symmetric stretching. The band at 973 cm$^{-1}$ was attributed to surface Si-OH stretching and transverse optical (TO) stretching of the silica network[23]. From the high-resolution XPS and Raman spectroscopy results, we infer that beyond mere adsorption, Si was integrated into the BaCaO$_2$ framework and formed Si-O-Ca/Ba-O-Si bonds. In addition, Co was covalently attached to the surface Si-OH groups, anchoring BCS to the perovskite surface.

To understand the impact of the interstitial oxygen concentration (δ) on the OER catalytic activity of BCS-PBCC, we undertook a comprehensive study of electrochemical oxygen intercalation and the oxygen diffusion rate, as illustrated in Fig. S17a. The peaks in the cyclic voltammograms (CV) represented insertion and extraction of the oxygen ions into and out of the lattice of the BCS-PBCC oxide, respectively, and the heights of these peaks indicated the rate of electron transfer and the mobility of oxygen ions in the perovskite[24]. In addition, the oxygen ion diffusion coefficients (DC) for both BCS-PBCC and PBCC were calculated through chronoamperometry, with the results are shown in the inset of Fig. S17b. BCS-PBCC was found to have a faster oxygen ion diffusion coefficient of 5.16 × 10$^{-13}$ cm$^2$ s$^{-1}$ at room temperature, which was ~7.2 times greater than that of PBCC. This suggested that an appropriate concentration of interstitial oxygen enhanced the electron transfer rate of the perovskite oxides and thus improved their catalytic activity[25]. The improved oxygen ion diffusion in BCS-PBCC is believed to be due to its unique layered structure, which was facilitated by the presence of A-site defects due to the increased oxygen vacancies.

## Important evidence for lattice oxygen participation

Our experimental findings revealed that the OER activities of the BCS-PBCC samples were markedly enhanced as the pH was raised from 12 to 14 (Fig. 3e). This direct relationship between the OER activity and pH indicated that the OER kinetics were governed by the LOM mechanism. In addition, we observed a more pronounced correlation between the catalyst activity and the pH as the silicon content was increased (Fig. 3f). This indicated that BCS functioned

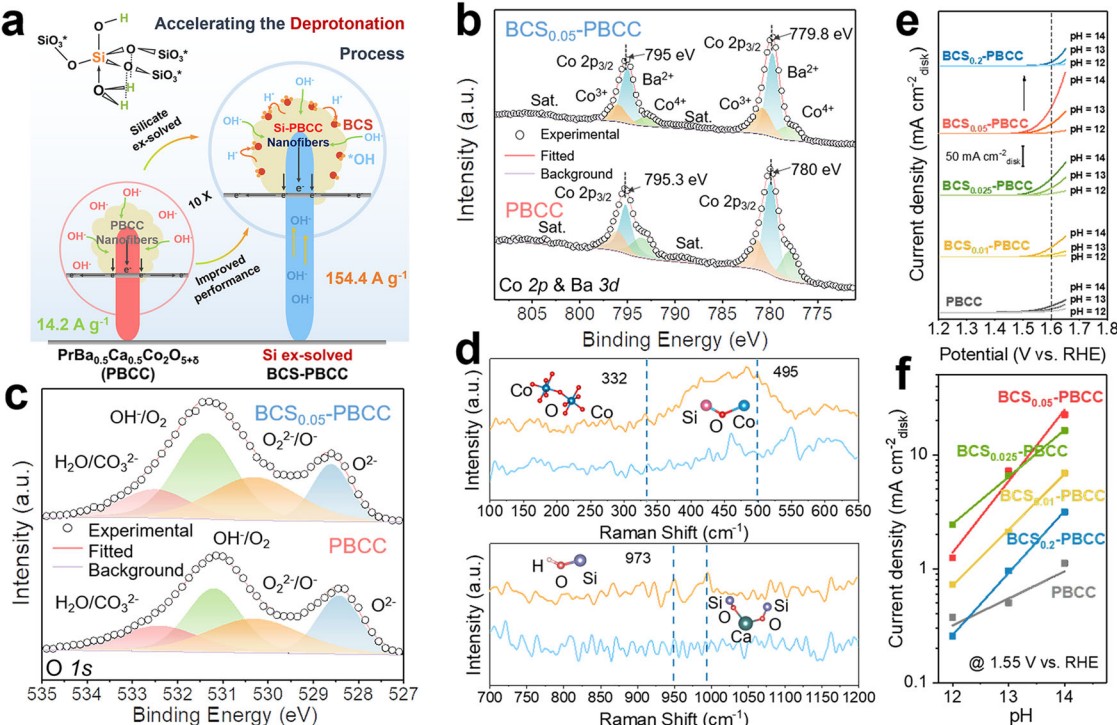

**Fig. 3 | Chemical and electrochemical characterization of the solid-hydroxy silicon/perovskite oxide. a** Structural formula of the hydroxy silicon and a schematic diagram showing the deprotonation process on the PBCC surface. **b** Co 2p core-level XPS spectra of $BCS_{0.05}$-PBCC and PBCC, with peak fitting results based on multiple cobalt species. Here "sat." denotes satellite peaks. **c** O 1 s core-level XPS spectra of $BCS_{0.05}$-PBCC and PBCC. **d** Raman spectra of BCS-PBCC (yellow) and PBCC (blue) samples. **e** OER kinetic currents (in mA $cm^{-2}_{geo}$) of various BCS-PBCC and PBCC samples in $O_2$-saturated KOH electrolytes with varying pH. **f** OER specific activities (in mA $cm^{-2}_{oxide}$) of various BCS-PBCC and PBCC samples at 1.55 V vs. RHE as a function of pH.

as a proton acceptor and accelerated the rate-limiting step in the OER.

To gain deeper insight into the LOM pathway and clarify the source of oxygen during the OER, we conducted DEMS measurements with the $^{18}O$ isotope. Our aim was to investigate the role of lattice oxygen in the OER (Fig. 4a)[26]. We first labeled the electrocatalyst with $^{18}O$ in a solution of 1.0 M KOH containing $H_2^{18}O$ and then rinsed it with $H_2^{16}O$ after the labeling process (Fig. S18a–c). We then tested the labeled catalyst in 1.0 M KOH containing $H_2^{16}O$ and used in situ trace DEMS to measure oxygen generation during the OER. The amount of $^{34}O_2$ released reflected the participation of lattice oxygens, which was quantified by OER cycling directly in the $^{18}O$ electrolyte (Fig. 4b)[27].

Our results confirmed that the lattice oxygens of BCS-PBCC were activated and released from the lattice matrix. Significantly, we discovered that only half of the oxygen atoms in the oxygen product came from the lattice, and the other half came from the electrolyte. This was also observed after several consecutive CV cycles of the catalyst, but the peak intensity for $^{18}O^{16}O$ appeared to weaken after several cycles (Fig. 4c), indicating that the $^{18}O$-labeled lattice oxygen was gradually consumed and replenished by $^{16}O$ from the electrolyte. The presence of trace amounts of $^{36}O$ confirmed that the OH* on the hydroxy silicon came from the $^{18}O$ in the electrolyte and participates in the entire OER process (Fig. 4d).

### DFT investigations of the deprotonation process
Density functional theory (DFT) calculations were performed to understand the reaction mechanism and proton transfer pathway of the BCS-PBCC system during the OER process (Fig. 5a, b). A comparison of the charge densities of the BCS and PBCC (Fig. 5c) and the BCS-PBCC heterostructure provided insight into the charge transfer mechanism. The Co atoms in the perovskite parent had higher negative charges (navy blue), while the Si atoms had higher positive charges

(light blue). This indicated electron transfer from the proton acceptor, BCS, toward the Co sites in PBCC. This flow facilitated cleavage of the O-H bonds of the intermediate OH* adsorbed on PBCC. This was attributed to the augmented electron density on the oxygen atoms within the $Si-O-Ca^{2+}/Ba^{2+}-O-Si$ configuration, which in turn increased their tendency to for H bonds[28]. Furthermore, we studied the electrochemical stages of the OER within the BCS-PBCC heterostructure. This process typically encompasses five foundational steps spanning both the AEM and LOM routes, which include four electrochemical steps and a singular nonelectrochemical step involving $O_2$ desorption (Fig. 5d).

As indicated by the experimental results, the BCS-PBCC system followed the LOM pathway. The formation of *OOH (where the asterisk indicates the active site on the catalyst surface) in step 2 serves was a potentially rate-determining step (PDS; 1.58 eV) in the LOM pathway, which was higher than in the AEM pathway and consistent with previous studies[18,29,30]. As the PDS switched from the formation of *OOH to the deprotonation of *OH, the strong proportional relationship between ΔG (*OOH) = G (*OH) + 3.2 eV in the AEM pathway became apparent. In the AEM, high catalyst activity is achieved by adjusting the binding strengths of the reaction intermediates on the surface, but it is limited by the linear scaling relationship and the charging behavior of the zinc ions caused by transition metal valence changes. A lattice oxygen-mediated mechanism has been proposed to speed up direct connection of the lattice oxygens and lower the limiting energy barrier. It has been reported that the mixed LOM pathway overcomes the proportional relationship between *OOH and *OH in the AEM pathway with a transition metal as the catalytic site, potentially leading to optimal activity when two neighboring oxidized oxygens can hybridize their oxygen vacancies without significantly sacrificing metal-oxygen hybridization (Figs. S19–22)[28,31]. Based on this, we propose a hybrid LOM in the presence of the proton acceptor-perovskite

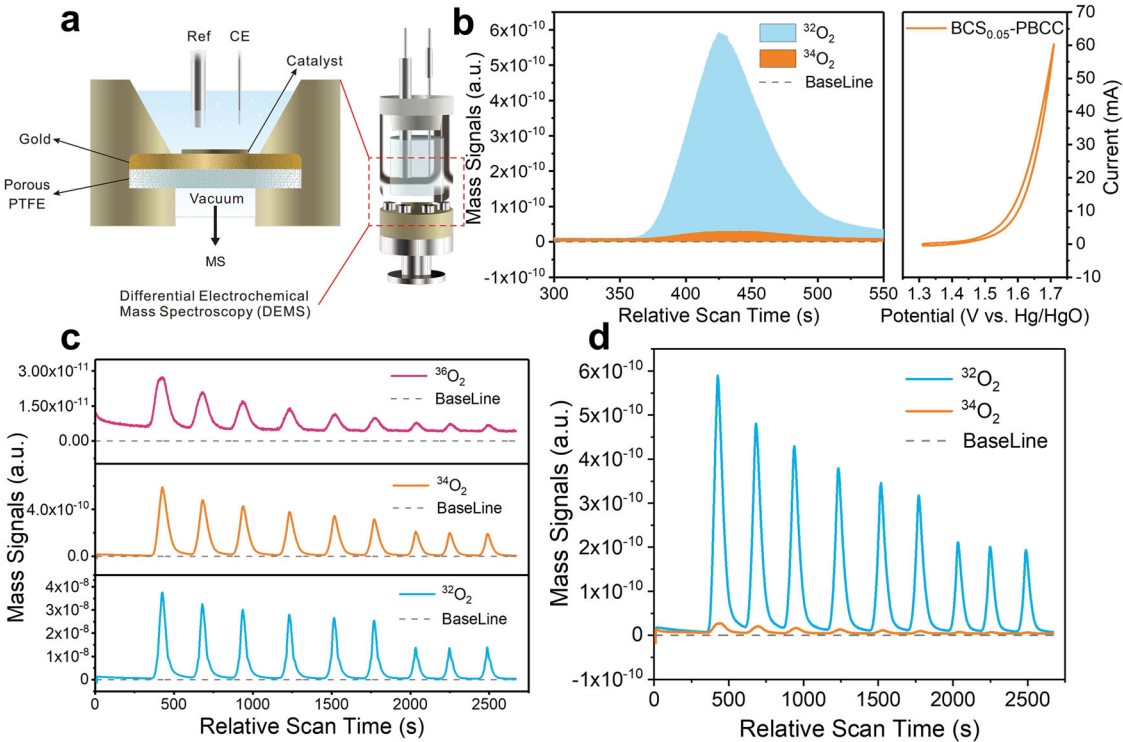

**Fig. 4 | DEMS signals for $^{34}O_2$ and $^{36}O_2$ in the reaction products formed by cycling in an $H_2^{18}O$ aqueous KOH electrolyte. a** Electrochemical flow cells for the differential electrochemical mass spectrometry (DEMS) setup and the DEMS dual thin-layer electrolyte flow cell (MS: mass spectrometry). **b** DEMS signals for $^{32}O_2$ ($^{16}O^{16}O$) and $^{34}O_2$ ($^{16}O^{18}O$) from the reaction products for the $^{18}O$-labeled BCS-PBCC catalyst in $H_2^{16}O$ aqueous KOH electrolyte and the corresponding CV cycles. The mass spectrometry signals were baseline-subtracted. **c, d** Plots of the DEMS signals versus time for $^{32}O_2$ ($^{16}O^{16}O$), $^{34}O_2$ ($^{16}O^{18}O$), and $^{36}O_2$ ($^{18}O^{18}O$) in the reaction products.

heterostructure (Fig. 5e). The relative stability and ΔG (activation free energy) for the two isomeric intermediates M1 (representing the Co adsorption sites in PBCC) and M2 (representing the Si adsorption sites in BCS) are key to correlating the OER mechanism with the different local configurations. Figure 5b illustrates the electron and proton transfers between M1 and M2 in the O-Co(OH)-O and O-Si-OH heterostructures as a starting point. Based on the above analysis, we believe that the BCS component facilitates deprotonation of the intermediate OH* to O*, thereby improving the overall kinetics of the OER in the composite system. The energetics of the rate-determining step (RDS) (OH* → O* + H*), which is deprotonation of the OH* to form O* + H*, are presented in Fig. 5f. In the BCS-PBCC system, the proton is adsorbed by the BCS, and the energy of the intermediate state O* + H* is reduced by −0.46 eV, making it much more accessible compared to deprotonation directly at the Co sites. On the other hand, PBCC has a higher ΔG for formation of O* + H*. In addition, we calculated the Gibbs free energies for different oxygen adsorption sites in the asymmetric structure, which revealed the optimal path for the OER process (Fig. 5g–i). Given that the distance between the O 2p band center and the Fermi level (EF) was regarded an important parameter to identify the activity of the lattice oxygen, we calculated the O 2p band center for BCS-PBCC and PBCC, and the values were −2.73 eV and −2.86 eV, respectively (Fig. S23). The O 2p band center after introducing BCS shifts towards EF, promoting the release of the lattice oxygen from the lattice, which will facilitate the LOM. This demonstrated the ability of the BCS-PBCC heterointerface to facilitate the deprotonation of OH* overcome the kinetic limits of the RDS.

## Charging test of the Zn-air battery

A Zn-air battery was fabricated with a gas diffusion layer coated with a mixture of BCS-PBCC and a Pt/C catalyst as the air electrode, a zinc foil as the anode, and 6.0 M KOH and 0.2 M $ZnCl_2$ aqueous solution as the

electrolyte (Fig. 6a, Supporting Information). For comparison, a rechargeable Zn-air battery was also fabricated with a mixture of $RuO_2$ and Pt/C as the air cathode. The charging rates of the above two air cathodes at 5–50 mA cm$^{-2}$ were first tested, as shown in Fig. 6b. At low currents, the voltages of the BCS-PBCC + Pt/C electrode and the $RuO_2$ + Pt/C electrode were similar, but with increasing currents, the charging voltage of the $RuO_2$ + Pt/C electrode increased sharply, while it increased smoothly for the BCS-PBCC + Pt/C electrode. The charging overpotential of the Zn-air battery with the BCS-PBCC + Pt/C electrode was just 0.30 V, which was much lower than the 0.42 V overpotential of the $RuO_2$ + Pt/C electrode. This suggested that BCS-PBCC achieved fast charge transfer with the electrolyte by overcoming the limitations of the RDS, thus resulting in faster charging of the Zn-air batteries[32,33].

The electrochemical stability of the Zn-air battery was also assessed with a galvanostatic charge/discharge test, as shown in Fig. 6c and S24. The BCS-PBCC + Pt/C electrode cycled stably for over 150 h, while state-of-the-art $RuO_2$ + Pt/C electrode cycled for just 80 h. In particular, the BCS-PBCC + Pt/C electrode exhibited first discharge and charge potentials of 1.24 V and 1.93 V, respectively. The charge/discharge voltage polarization was only 0.69 V, and the energy efficiency was up to 64.2%. In contrast, the charge/discharge voltage polarization and energy efficiency were just 0.76 V and 61.9% for the commercial mixed catalyst $RuO_2$ + Pt/C. After cycling for 150 hours, the charge polarization voltage of the BCS-PBCC + Pt/C electrode had increased by only 20 mV, which was much lower than the value of 340 mV for $RuO_2$ + Pt/C after 80 h (Fig. 6d). In addition, a well-defined recharging platform was observed for BCS-PBCC through all cycles, which confirmed that only the OER occurred during the charging process. These results confirmed that BCS-PBCC is a robust OER catalyst for recharging Zn-air cells with heteroatom-doped carbon materials used to prepare a noble metal-free bifunctional electrocatalyst[34–38]. In addition, given that BCS-PBCC also served as the anode for water electrolysis, we

propose the concept of using Zn-air batteries to drive a water electrolyzer, as shown in Fig. 6e. As a proof of concept, the water electrolyzer with BCS-PBCC || Pt/C showed a lower overpotential than that for RuO2 || Pt/C, especially at a high current density (Fig. 6f).

## Discussion

In this work, we present a unique approach to enhancing the OER activity by introducing BCS as a proton acceptor onto the surface of PBCC via a simple one-step exsolution process. The resulting BCS-

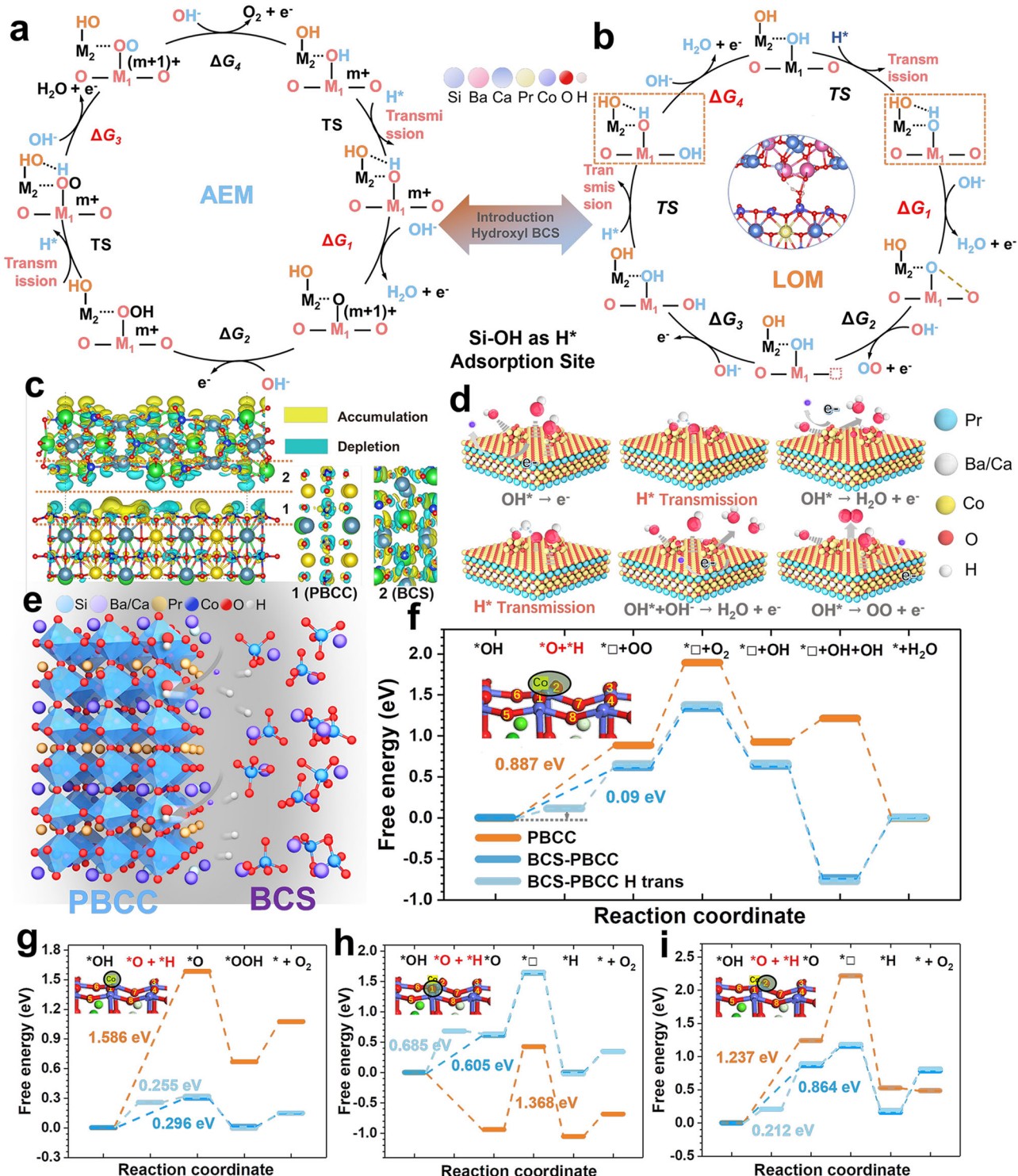

**Fig. 5 | DFT calculations and the electrocatalysis mechanism.** Proposed OER mechanisms, including **a** AEM and **b** LOM. M1 (Co) and M2 (Si) are the adsorption centers of the AEM and LOM in the OER pathway. Other intermediates are also labeled. The empty square represents an oxygen vacancy. **c** Charge density difference plot for the interface between BaCaSiO4 and PrBa0.5Ca0.5Co2O5+δ. A side view of the total structure (left) and the cross-sections of the layers of the PBCC (middle) and BCS (right) are shown. Color representation: Pr (green), Ba/Ca (gray), Co (navy blue), O (red), Si (light blue). **d** Adsorption/desorption and electron transfer processes of OER intermediates on the surface of BCS-PBCC. **e** Schematic diagram of the deprotonation steps at the interface. **f** Free energies of the LOM and **g** AEM reaction pathways on PBCC and hydroxy silicon BCS-PBCC. **h** Free energies of the LOM via noncooperative proton-electron transfer steps involving different configurational lattice oxygens (**h** O1 and **i** O2). **f**–**i** Color representation: Pr (green), Ba/Ca (gray), Co (purple), O(red).

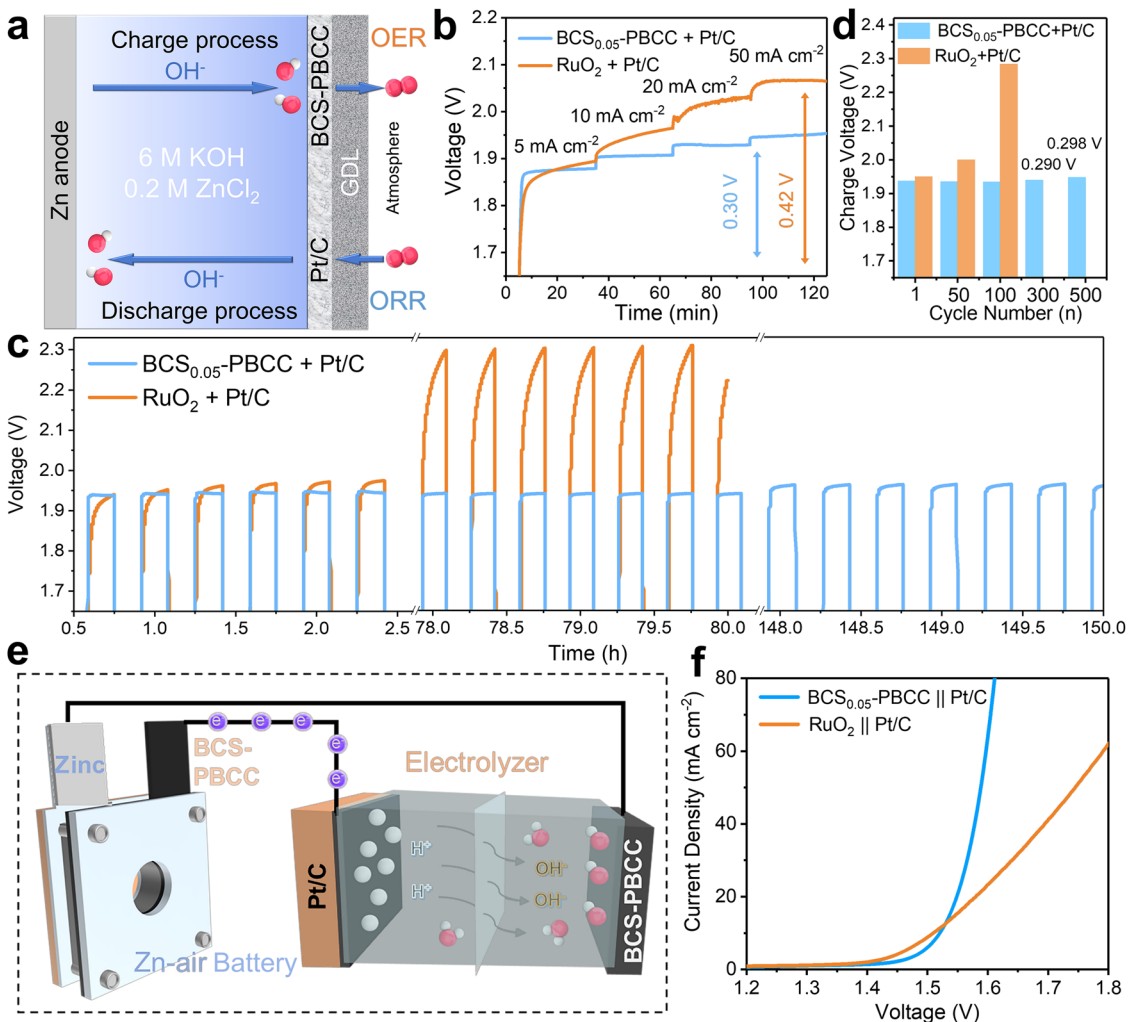

**Fig. 6 | Recharging performance of the Zn-air battery. a** Schematic diagram for recharging of the Zn-air battery. **b** Galvanostatic charging tests at different current densities. **c** Galvanostatic charge/discharge test at 5 mA cm⁻², **d** and the corresponding charging potentials at different cycle stages. **e** A Zn-air battery was used as the power source to drive the water electrolysis device, in which the BCS-PBCC catalyst was used as both the cathode for the Zn-air battery and the anode for water electrolysis. **f** Polarization curves for water electrolysis in 1.0 M KOH. Color representation in (**a**) and (**e**): O (red), H (gray).

PBCC demonstrated a low OER overpotential (300 mV) with an onset potential of 1.45 V at 10 mA cm⁻² in 1.0 M KOH. The amount of Si incorporated significantly affected the Tafel slope, which decreased as the amount of Si increased; this indicated that the BCS overcame the deprotonation barrier in the OER process. DFT calculations revealed a mechanism involving both lattice oxygen participation and metal site adsorption in the BCS-PBCC heterostructure. The unique structure (OH-Si-O-Ca²⁺/Ba²⁺-O-Si-OH) of BCS at the M2 sites greatly accelerated the OH* deprotonation process on the perovskite (M1 site) surface, reducing the Gibbs free energy of the proton transfer transition state. A rechargeable Zn-air battery using BCS-PBCC exhibited a stable charging voltage for 150 hours owing to the customized reaction mechanism that avoided zinc-ion battery behavior. This study presents a scalable approach for developing highly efficient OER electrocatalysts with proton acceptor configurations and opens new avenues for designing efficient OER electrocatalysts for rechargeable Zn-air batteries and water-splitting devices.

## Methods

### Materials syntheses
The BCS-PBCC perovskite samples were prepared by electrospinning. Analytical grade raw materials were directly employed without any additional purification. Accordingly, stoichiometric amounts of

Pr(NO₃)₃·4H₂O, Ba(NO₃)₂, Ca(NO₃)₂·4H₂O, Co(NO₃)₂·6H₂O and tetraethyl orthosilicate (C₈H₁₂O₈Si) were dissolved in 12 mL of N,N-dimethylformamide (DMF). Then, 1 g of polyacrylonitrile (PAN) was gradually added into the precursor solution after all of the metal nitrates were dissolved by immersion in a water bath at 80 °C. The mixture was stirred at a controlled temperature of 80 °C for 20 h to ensure that the nitrate mixture and PAN were completely dissolved and homogenized. The prepared viscous precursor gel was filled into a plastic syringe with a stainless-steel needle. The plastic syringe was fixed to an electrostatic spinning machine and then the high-voltage power supply was adjusted to ensure the formation of fibers through electrostatic force and deposition on the tin foil of a synchronously rotating metal drum. The obtained precursor was dried and calcined at 400 °C for 6 h. Finally, the catalysts, including PBCC, BCS₀.₀₁-PBCC, BCS₀.₀₂₅-PBCC, BCS₀.₀₅-PBCC, and BCS₀.₂-PBCC nanofibers with different tetraethyl orthosilicate contents in the precursor, were obtained after calcining at 1000–1100 °C for 24 h in air.

### Basic characterizations
The crystal structures and purities of the pristine perovskite PBCC and BCS-PBCC powder were determined by XRD with Cu Kα irradiation (D8 Advance, Bruker diffractometer) at a scan speed of

$2° \text{ min}^{-1}$ and a scan range of 20–80°. The microstructures were observed with a scanning electron microscope, and the lattice structures were analyzed by transmission electron microscopy using high-resolution TEM (JEOL JEM-2100F). The element distributions were determined with energy-dispersive spectroscopy. The Brunauer–Emmett–Teller (BET) specific surface areas of the catalysts were determined with $N_2$ adsorption/desorption and an Autosorb Quantachrome 1 MP system. The $O_2$-TPD tests were conducted with a TP-5080 system. X-ray photoelectron spectroscopy (XPS) was performed with a Kratos AXIS UltraDLD-600 W XPS system outfitted with an Al Kα X-ray source (1468.6 eV). All data were normalized relative to the C 1s binding energy of 284.7 eV.

## In situ electrochemical mass spectrometry

The $^{18}O$ labeling process was used with a differential electrochemical mass spectrometer (DEMS, Model: QAS100). The catalyst ink was prepared by dispersing 2.8 mg of BCS-PBCC into a mixed solution with 500 µL of ultrapure water, 460 µL of ethanol, and 40 µL of Nafion and ultrasonication for 1 h. Then, 20 µL of the as-prepared ink was drip-coated on the gold film of the in situ electrolytic cell and dried naturally for further testing. For the $^{18}O$ labeling studies in 1.0 M KOH electrolyte, a carbon rod and Hg/HgO were used as the counter and reference electrodes, respectively. For LSV and CV tests, the potential ranges were 0.5–0.9 V with a scanning rate of $5 \text{ mV s}^{-1}$.

## Electrochemical measurements

The electrochemical performance was detected with an electrochemical workstation (AUT87986, Metrohm Autolab B. V., Netherlands) and a three-electrode configuration, in which saturated Hg/HgO and a carbon rod were employed as the reference and counter electrodes, respectively, in 0.1 M and 1.0 M KOH electrolyte. $BCS_x$-PBCC ($x = 0.01, 0.025, 0.05, 0.2$) and PBCC catalyst inks were made by dispersing 6 mg of catalyst and 3 mg of acetylene black into a mixed solution containing 50 µL Nafion (5 wt.%), a small amount of N,N-dimethylformamide (DMF), and 100 µL ultrapure water with sonication. Then, 3.5 µL of the catalyst inks were dripped onto a glassy carbon disk electrode with an area of $0.1256 \text{ cm}^2$ to create the working electrode with a loading of $0.202 \text{ mg cm}^{-2}$. CV cycling activation was needed prior to testing. Specifically, the tested sample was cycled 30 times over the voltage range −0.7 to 0.1 V. The LSV polarization curves were measured from 1.0 to 2.0 V (vs. RHE) in an $O_2$-saturated electrolyte at a scan rate of $5 \text{ mV s}^{-1}$. The electrochemical tests were performed at an ambient temperature of 25 °C.

Using the Nernst equation, the potentials were calculated with the Formula $E = E_0 − iR + 0.92$ for 1.0 M KOH (pH 14) based on the Hg/HgO reference electrode. Electrochemical impedance spectroscopy (EIS) measurements were performed over the frequency range 100 kHz to 0.01 Hz with an AC voltage, an overpotential of 300 mV and an amplitude of 10 mV.

## Turnover frequencies (TOFs) for the OER

Equation (1) was used to compute the TOFs for the OER with each catalyst to evaluate the capacity of the catalytic sites:

$$TOF = JA/4Fn \tag{1}$$

where J, A, F, and n are the current density at 1.56 V for the OER (mA $\text{cm}^{-2}$), electrode area ($\text{cm}^2$), Faraday constant (96,500 A s $\text{mol}^{-1}$), and the amount of catalyst (moles), respectively. We used the Supplementary Equation (2):

$$TOF = \frac{\#\text{number of total oxide turnovers/cm}^2}{\#\text{number of active sites/cm}^2} \tag{2}$$

## Calculation of the specific activity (SA) and mass activity (MA)

The specific activity (mA $\text{cm}^{-2}_{ox}$) was calculated with the equation:

$$\text{Specific activity} = \frac{J}{10 * m * S_{BET}} \tag{3}$$

The mass activity (A $\text{g}^{-1}_{metal}$) was derived from the current density (mA $\text{cm}^{-2}$) and was normalized with the mass loading of the metal at a certain applied overpotential. The following equation was used to calculate the mass activity of BCS-PBCC:

$$\text{Mass activity} = \frac{|J|}{m} = 431.03|J|\text{A g}_{metal}^{-1}\text{per cm}^2 \tag{4}$$

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

## Acknowledgements

This research was supported by the National Natural Science Foundation of China (22272081) and the Jiangsu Provincial Specially Appointed Professors Foundation and the Graduate Student Scientific Research Innovation Projects in Jiangsu Province (KYCX22_1186). C.Y. acknowledges support from the Guangdong Basic and Applied Basic Research Foundation (2022A1515110628). Computing resources were supported by the Center for Computational Science and Engineering at Southern University of Science and Technology.

## Author contributions

Y.F.B. and Z.P.S. conceived and designed the research. Y.B.W. and Q.L. conducted characterizations and electrochemical measurements. Y.B.W. and Y.F.B. were involved in the structural and electrochemical analysis. W.J.B., C.C.Y., and X.L.G. were involved in the DFT calculation analysis. All authors discussed and analyzed the data. Y.B.W., Q.L., and Y.F.B. cowrote the manuscript.

## Competing interests

The authors declare no competing interests.
