## [Peer Review File · Nature Communications]

Accelerated deprotonation with a hydroxy-silicon alkali solid for rechargeable zinc-air batteriesReviewers' Comments:

Reviewer #1:

Remarks to the Author:

Authors report OER catalyst research based on the perovskite PBCC oxide catalyst with BCS surface decoration as the main part of the reported work. Authors also report an application of BCS+PBCC catalyst mixed with Pt/C in Zn-air battery cathode in a rather brief study summarized in Fig. 6.

The most prominent results articulated by the authors is that BCS+PBCC catalyst has superior OER performance than PBCC catalyst: overpotential of 300 mV vs. 415 mV at 10 mA/cm² current, and Tafel slope of 49 mV/dec vs. 85 mV/dec. Most discussions in Figs. 3-5 are focused on explaining the nature and origin of this improvement of BCS+PBCC over PBCC as OER catalyst. Even though this finding is technically interesting, the OER performance itself is not as remarkable. Specifically, PBCC catalyst data would place it as the worst catalyst compared in Fig. 2f. PBCC with Tafel slope = 85 mV/dec and overpotential = 415 mV will place it at the upper right corner of the chart as one of the worst catalyst. The addition of BCS bring the BCS+PBCC catalyst down to lower left corner comparable to ScCo(0.85)Fe(0.1)P(0.05)O(3-d) and LaCo(0.8)V(0.2)O₃. In fact these other catalysts have superior overpotential or Tafel slope to BCS+PBCC catalyst. In summary, BCS addition can improve the low quality catalyst OER performance comparable to those of high performance OER catalysts.

Based on this analysis, I would recommend the authors to submit the manuscript to technically focused journal.

Reviewer #2:

Remarks to the Author:

In the present work, the authors integrated a stable solid base, hydroxyl BaCaSiO₄ (BCS), into the surface of PrBa_{0.5}Ca_{0.5}Co₂O_{5+δ} (PBCC) perovskite. And the presence of the HO-Si site on the hydroxyl BCS significantly accelerates proton transfer from adsorbed OH* on the PBCC, and thus the OER catalytic activity has been improved immensely. I recommend that the work could be published in Nature Communications, after addressing the following points:

1. How about the ORR performance from the composite catalysts? This is because Pt/C was used as the ORR catalysts in the battery test.
2. The mechanism in the composite catalysts follows the mixed AEM and LOM. The authors should clarify or explain that is there mutual influence in each other? This is because when lattice oxygen was activated, the electronic structure would be altered, and thus the performance following AEM could be altered correspondingly.
3. The hydrogen transmission should be proved in the revised manuscript.
4. The Tafel analysis should be performed in the similar current density range.
5. Please use high-definition images in the revised manuscript.

Reviewer #3:

Remarks to the Author:

This manuscript demonstrates that the presence of the HO-Si site facilitates the proton transfer of the adsorbed OH* on the active site. The authors integrated stable hydroxyl BaCaSiO₄ as a proton acceptor onto the surface of PrBa_{0.5}Ca_{0.5}Co₂O_{5+δ} perovskite nanofibers via a one-step calcination strategy. They proposed a novel deprotonation mechanism to solve the charging behavior of zinc ions affected by the change of transition metal valence in rechargeable zinc-air batteries.

This work is very interesting. The proton transfer process on the oxide surface is resolved by in-situ characterization, and the introduction of inert proton acceptors clearly explains the key issue of the source of activity. The mechanism of oxygen evolution involving hydroxyl silicon and lattice oxygen proposed by the authors provides a new platform for the design of catalysts for

rechargeable zinc-air batteries. The quality of the data in the manuscript is very good and the measurements are very reliable.

It is recommended that a slightly revised version be published in Nature Communications.

Here are my concerns:

1. Maintaining consistent line colors in Figures 2a and 2b is recommended.
2. Why does the electrochemical surface area (ECSA) increase after integrating BCS particles?
3. In the experimental section, you mentioned the BET test information, but there are no test results of BET in the manuscript.
4. Can Pr dope into the lattice of BCS after calcination?
5. The specific process of deprotonation in the new mechanism should be described in detail, which is very important for understanding the evolution of protons on the catalyst surface.
6. References are very weak. Several references deal with the effect of proton acceptors on catalysts and need to be considered.
7. Comparisons with other proton acceptor catalysts are lacking in this manuscript, especially in the last two years. This is very important.
8. Carefully confirm that all acronyms are defined when they first appear in the abstract and manuscript. Only extended forms are allowed if the element is referenced only once in the article.
9. You need to cite the most relevant literature of your current manuscript, for example, the first paper of Ca doped double perovskite (NBCO). "Development of double-perovskite compounds as cathode materials for low temperature solid oxide fuel cell" *Angewandete Chem. Int. Ed*, 2014, 53, 13064-13067.

Response Letter

Referee:

Reviewer #1: Comments to the Author

Authors report OER catalyst research based on the perovskite PBCC oxide catalyst with BCS surface decoration as the main part of the reported work. Authors also report an application of BCS+PBCC catalyst mixed with Pt/C in Zn-air battery cathode in a rather brief study summarized in Fig. 6.

The most prominent results articulated by the authors is that BCS+PBCC catalyst has superior OER performance than PBCC catalyst: overpotential of 300 mV vs. 415 mV at 10 mA/cm² current, and Tafel slope of 49 mV/dec vs. 85 mV/dec. Most discussions in Figs.3-5 are focused on explaining the nature and origin of this improvement of BCS+PBCC over PBCC as OER catalyst. Even though this finding is technically interesting, the OER performance itself is not as remarkable. Specifically, PBCC catalyst data would place it as the worst catalyst compared in Fig. 2f. PBCC with Tafel slope = 85 mV/dec and overpotential = 415 mV will place it at the upper right corner of the chart as one of the worst catalyst. The addition of BCS bring the BCS+PBCC catalyst down to lower left corner comparable to ScCo_(0.85)Fe_(0.1)P_(0.05)O_(3-δ) and LaCo_(0.8)V_(0.2)O₃. In fact these other catalysts have superior overpotential or Tafel slope to BCS+PBCC catalyst. In summary, BCS addition can improve the low quality catalyst OER performance comparable to those of high performance OER catalysts.

Based on this analysis, I would recommend the authors to submit the manuscript to technically focused journal.

The author's answer: We are grateful to the reviewer for the time and effort spent on reviewing this manuscript. Our work is mainly devoted to developing high-efficiency oxygen catalysts for zinc-air batteries by customizing reaction mechanisms. Zinc-air battery using transition metal oxide catalysts always encounter the zinc-ion battery behavior caused by valence change of transition metal, which can affect the energy

efficiency of the redox reaction of O_2/OH^- . This zinc-ion battery behavior was mainly caused by the conventional adsorbate evolution mechanism. Therefore, we propose a new lattice oxygen mechanism to avoid this issue of rechargeable zinc-air batteries. Our work designs a composite catalyst by introducing a proton acceptor to break the deprotonation that is regarded as the rate-limiting step of the lattice oxygen mechanism. The reaction mechanism was well clarified by using in-situ differential electrochemical mass spectrometry and density functional theory calculations. More importantly, the advantages of the proposed lattice oxygen mechanism are validated in zinc-air batteries, which achieve a stable charging voltage platform for a long time without zinc-ion battery behavior. We believe our work is highly innovative in material design, mechanism explanation, and theoretical guidance for achieving high-performance of rechargeable zinc-air batteries.

In order to highlight the innovations of this work, we also revised the abstract and introduction. and highlighted with yellow background in the revised manuscript, as follows:

Abstract revision:

Transition metal oxide are promising electrocatalyst for Zn-air batteries, yet their surface reconstruction caused by the adsorbate evolution mechanism (AEM), which induce the zinc-ion batteries behavior, in the oxygen evolution reaction (OER) leads to poor cycle performance. In this study, we propose a new lattice oxygen mechanism (LOM) involving proton transfer to solve the issue of valence state changes of transition metals in electrocatalysts during OER process. Here we choose perovskite oxide as the research model by introducing a stable solid base, hydroxyl BaCaSiO₄, onto the surface of PrBa_{0.5}Ca_{0.5}Co₂O_{5+δ} perovskite nanofibers using a one-step exsolution strategy. The HO-Si site on the hydroxyl BaCaSiO₄ significantly accelerates proton transfer from OH* adsorbed on PrBa_{0.5}Ca_{0.5}Co₂O_{5+δ}. We further propose a novel deprotonation mechanism involving lattice oxygen and hydroxyl silicon for promoting OER kinetics based on the theoretical and experimental results. As a proof of concept, the recharging Zn-air batteries assembled with this composite electrocatalyst achieve a stable charge and discharge voltage platform for over 150 hours. Our findings open new avenues for

designing efficient OER electrocatalysts for rechargeable Zn-air batteries.

Introduction revision:

Rechargeable Zn-air batteries have received great attention owing to their high energy density, environmentally friendly, and low cost, in which oxygen evolution reaction (OER) determines the charging efficiency.¹⁻⁴ Recently, developing efficient oxygen electrocatalysts is considered the hinge to realize the practical Zn-air battery. Transition metal oxide is an important branch of oxygen electrocatalysts. However, the conventional adsorbate evolution mechanism (AEM) would induce the valence state change of transition metal and even surface reconstruction during OER, thus causing the zinc-ion batteries behavior, in detail, occurring two voltage platforms. To track this issue, the precise design of advanced oxygen electrocatalysts is imperative for rechargeable Zn-air batteries.

Recently, the lattice oxygen mechanism (LOM) was proposed to break the linear relationship and limit of theoretical overpotential for AEM. As for LOM, the catalytic site is the oxygen site, while the transition metal not involves in the reaction. Given this, we believe that introducing LOM into Zn-air batteries can avoid the zinc-ion batteries behavior. However, OER following LOM encountered poor stability and a high proton desorption kinetic barrier, thus it is necessary to modify oxygen electrocatalysts that follow LOM to improve their activity and stability.

Reviewer #2: Comments to the Author

In the present work, the authors integrated a stable solid base, hydroxyl BaCaSiO₄ (BCS), into the surface of PrBa_{0.5}Ca_{0.5}Co₂O_{5+δ} (PBCC) perovskite. And the presence of the HO-Si site on the hydroxyl BCS significantly accelerates proton transfer from adsorbed OH* on the PBCC, and thus the OER catalytic activity has been improved immensely. I recommend that the work could be published in Nature Communications, after addressing the following points:

- 1. How about the ORR performance from the composite catalysts? This is because Pt/C was used as the ORR catalysts in the battery test.**

The author's answer: Thank you for the useful comments on our manuscript. We have modified the manuscript accordingly, and detailed corrections are listed below point by point. The change is highlighted with yellow background. The point-by-point response is as follows: In electrochemical tests, we evaluated the ORR performance of BCS-PBCC and PBCC. However, in this study, we mainly confirmed the deprotonation process of BSC proton acceptor in OER and then optimized the charging performance in zinc-air batteries. To compare with commercial RuO₂, Pt/C was used as ORR catalysts in order to probe the charging performance of BCS-PBCC in zinc-air batteries. Based on your suggestion, the ORR performance is presented in the Supporting Information and highlighted with yellow background in the manuscript as follows:

Fig. S12. LSV curves of ORR for PBCC and BCS-PBCC.

2. The mechanism in the composite catalysts follows the mixed AEM and LOM. The authors should clarify or explain that is there mutual influence in each other? This is because when lattice oxygen was activated, the electronic structure would be altered, and thus the performance following AEM could be altered correspondingly.

The author's answer: According to a previous report (*Nat. Energy* 2019, 4 (4), 329–338), the switch of the OER mechanism from the AEM to the LOM occurs only if two neighboring oxygens can hybridize their oxygen vacancy but not sacrifice metal–oxygen hybridization significantly. In the process of AEM switching LOM, many directional Co–O covalent bonds are changed. Meanwhile, due to the strong scaling relation $G(*OOH) = G(*OH) + 3.2 \text{ eV}$ in the AEM path, the PDS switches from the formation of * OOH to the deprotonation of * OH on BCS-PBCC. BCS-PBCC shows a reduced OER overpotential according to the AEM calculation but is still very high for the whole OER process. Therefore, in this hybrid mechanism, BCS-PBCC will preferentially proceed through the LOM pathway.

3. The hydrogen transmission should be proved in the revised manuscript.

The author's answer: We apologize for not being able to directly observe the proton transfer process through experiments. However, we believe that current experimental and characterization results can well prove that BCS accelerates the deprotonation rate in PBCC system. According to the experimental results, the deprotonation process is the rate-determining step for PBCC, and the overpotential is significantly reduced after introducing BCS. In contrast, the physical mixture of BCS and PBCC shows slightly enhanced OER catalytic activities, which confirms that surface BCS can decrease the energy barrier of OER. Tafel slope and density functional theory calculations both prove that the deprotonation process is the rate-determine step of PBCC. Therefore, we think the surface BCS can facilitate the deprotonation rate during OER process, thus enhancing the catalytic activities.

4. The Tafel analysis should be performed in the similar current density range.

The author's answer: The data in Fig. 2b have been revised as your suggestion, and highlighted with yellow background in the revised manuscript as follows:

5. Please use high-definition images in the revised manuscript.

The author's answer: We have updated all the pictures as you requested and submitted them in additional files.

Reviewer #3: Comments to the Author

This manuscript demonstrates that the presence of the HO-Si site facilitates the proton transfer of the adsorbed OH* on the active site. The authors integrated stable hydroxyl BaCaSiO₄ as a proton acceptor onto the surface of PrBa_{0.5}Ca_{0.5}Co₂O_{5+δ} perovskite nanofibers via a one-step calcination strategy. They proposed a novel deprotonation mechanism to solve the charging behavior of zinc ions affected by the change of transition metal valence in rechargeable zinc-air batteries.

This work is very interesting. The proton transfer process on the oxide surface is resolved by in-situ characterization, and the introduction of inert proton acceptors clearly explains the key issue of the source of activity. The mechanism of oxygen evolution involving hydroxyl silicon and lattice oxygen proposed by the authors provides a new platform for the design of catalysts for rechargeable zinc-air batteries. The quality of the data in the manuscript is very good and the measurements are very reliable.

It is recommended that a slightly revised version be published in Nature Communications.

Here are my concerns:

1. Maintaining consistent line colors in Figures 2a and 2b is recommended.

The author's answer: We thank the reviewer for the considerate comment. We have modified the manuscript accordingly, and detailed corrections are listed below point by point. The change is highlighted with yellow background. We took your comments seriously and tried to unify the line colors in Fig. 2. However, due to the overlap of lines in Fig. 2b, we used easily distinguishable line colors and provided a detailed legend explanation.

2. Why does the electrochemical surface area (ECSA) increase after integrating BCS particles?

The author's answer: The enhanced ECSA is mainly attributed to the increase in specific surface area from the results of the BET test as shown in Fig. S10. The exsolution of BCS to PBCC surface can improve the surface roughness of BCS-PBCC,

thus enhancing the specific surface area. In order to exclude the influence of specific surface area on performance, we further compared the intrinsic activity, and highlighted with yellow background in the revised Supporting Information, as follows:

Fig. S12. The intrinsic activity of BCS-PBCC and PBCC.

3. In the experimental section, you mentioned the BET test information, but there are no test results of BET in the manuscript.

The author's answer: We have provided the BET results in the revised Supporting Information, and highlighted with yellow background, as follows:

Fig. S10. Nitrogen adsorption-desorption isotherm curves of PBCC and BCS-PBCC samples.

4. Can Pr dope into the lattice of BCS after calcination?

The author's answer: 4. Pr ions cannot be doped into the BCS lattice due to their large diameter. From the XRD result in Fig. S2, the diffraction peak of BCS-PBCC does not shift compared to PBCC, which proves that Pr ions are mainly doped into the lattice of PBCC. In addition, the refined XRD in Fig. 2c also confirms that the lattice parameters of BCS and PBCC are not affected.

5. The specific process of deprotonation in the new mechanism should be described in detail, which is very important for understanding the evolution of protons on the catalyst surface.

The author's answer: The O-Co (OH)-O and O-Si-OH interface structures were used as the unified starting point. In the OER evolution mechanism of BCS-PBCC, the proton is adsorbed by the BCS top, and the intermediate state $O^* + H^*$ is lowered by -0.46 eV, which is much easier than the direct deprotonation at the Co site. On the other hand, PBCC has a higher ΔG for $O^* + H^*$. In addition, we calculate the Gibbs free energy of different oxygen adsorption sites in the asymmetric structure, revealing the optimal path in the OER process. This demonstrates the ability of the BCS-PBCC heterointerface to promote the deprotonation of OH^* , breaking the kinetic limit of RDS.

6. References are very weak. Several references deal with the effect of proton acceptors on catalysts and need to be considered.

The author's answer: We have added references about the effect of proton acceptor on catalysts, highlighted with yellow background in the revised manuscript, as follows:

36. Bonin, J., Costentin, C., Louault, C., Robert, M. & Savéant, J. M. Water (in Water) as an intrinsically efficient proton acceptor in concerted proton electron transfers. *J. Am. Chem. Soc.* **133**, 6668–6674 (2011).

37. Coggins, M. K., Zhang, M.-T., Chen, Z., Song, N. & Meyer, T. J. Single-Site Copper(II) Water Oxidation Electrocatalysis: Rate Enhancements with HPO_4^{2-} as a Proton Acceptor at pH 8. *Angew. Chemie* **126**, 12422–12426 (2014).

7. Comparisons with other proton acceptor catalysts are lacking in this manuscript, especially in the last two years. This is very important.

The author's answer: We have added the comparison between our work and other proton acceptor catalysts in the revised supporting information and revised manuscript, and highlighted with yellow background as follows:

Proton Acceptor-Oxide Composite Catalyst						
PO_4^- $\text{PrBa}_{0.5}\text{Ca}_{0.5}\text{Co}_2\text{O}_{5+\delta}$	290	51	0.1 M KOH	0.202	GC	41
$\text{Sr}(\text{Co}_{0.8}\text{Fe}_{0.2})_{0.7}\text{B}_{0.3}\text{O}_{3-\delta}$	340	58	0.1 M KOH	0.232	GC	42
$\text{MoS}_2@\text{SrCoO}_{3-\delta}$	351	37	0.1M KOH	/	GC	43

8. Carefully confirm that all acronyms are defined when they first appear in the abstract and manuscript. Only extended forms are allowed if the element is referenced only once in the article.

The author's answer: We carefully examined and revised the acronyms and definitions throughout the manuscript, and highlighted with the yellow background in the revised manuscript.

9. You need to cite the most relevant literature of your current manuscript, for example, the first paper of Ca doped double perovskite (NBCO). "Development of double-perovskite compounds as cathode materials for low temperature solid oxide fuel cell" *Angewandete Chem. Int. Ed*, 2014, 53, 13064-13067.

The author's answer: We have added the relevant reference to improve the quality of our manuscript, highlighted with yellow background in the revised manuscript, as follows:

38. Yoo, S. *et al.* Development of Double-Perovskite Compounds as Cathode Materials for Low-Temperature Solid Oxide Fuel Cells. *Angew. Chemie* **126**, 13280–13283 (2014).

Reviewers' Comments:

Reviewer #2:

Remarks to the Author:

The authors have made revisions based on the existing feedback, and the following are additional modification suggestions.

1, The X-axis labels of Fig. 3a appear somewhat distorted, and there is an overlap between Fig. 3d and the image. Please carefully review the figures.

2, Would the acceleration of the OH* 374 deprotonation process have adverse effects on the negative electrode interface?

Reviewer #3:

Remarks to the Author:

The current form fits for the publication.

Thanks.

Referee:

Reviewer #2: Comments to the Author

1. The X-axis labels of Fig. 3a appear somewhat distorted, and there is an overlap between Fig. 3d and the image. Please carefully review the figures.

2. Would the acceleration of the OH* deprotonation process have adverse effects on the negative electrode interface?

The author's answer: Thank you for the useful comments on our manuscript. We have modified the manuscript accordingly, and detailed corrections are listed below point by point. The change is highlighted with yellow background. The point-by-point response is as follows:

1. We have corrected the distorted part and replotted the overlapping part in **Fig.3**. According to your suggestion, the revised image is highlighted in yellow in the manuscript, as follows:

Fig. 3. Chemical and electrochemical characterisations of solid-hydroxy silicon/perovskite oxide.

2. The constructed proton acceptor system mainly acts on the intermediate/target adsorbed on the interface, and this force can reduce the reaction barrier during the deprotonation process, which makes the H-O bond easier to break. In the mechanism diagram of **Fig.5**, the fourth step reaction describes this process in detail, and the accelerated desorption only accelerates the binding rate of H* in the adsorbed OH* and OH⁻ in the electrolyte. The protons combine with the OH⁻ in the electrolyte and then return to their original state, forming a cycle. Therefore, we believe that the effect of this deprotonation process on the cathode is almost negligible.